# Microbiota Regulates Pancreatic Cancer Carcinogenesis through Altered Immune Response

**DOI:** 10.3390/microorganisms11051240

**Published:** 2023-05-08

**Authors:** Yihan Chai, Zhengze Huang, Xuqiu Shen, Tianyu Lin, Yiyin Zhang, Xu Feng, Qijiang Mao, Yuelong Liang

**Affiliations:** 1Department of General Surgery, Zhejiang University School of Medicine, Sir Run Run Shaw Hospital, Hangzhou 310016, China; 2Zhejiang Provincial Key Laboratory of Laparoscopic Technology, Hangzhou 310016, China; 3Zhejiang Province Medical Research Center of Minimally Invasive Diagnosis and Treatment of Abdominal Diseases, Hangzhou 310028, China

**Keywords:** pancreatic cancer, microbiota, microbiome, immunomodulation, inflammation

## Abstract

The microbiota is present in many parts of the human body and plays essential roles. The most typical case is the occurrence and development of cancer. Pancreatic cancer (PC), one of the most aggressive and lethal types of cancer, has recently attracted the attention of researchers. Recent research has revealed that the microbiota regulates PC carcinogenesis via an altered immune response. Specifically, the microbiota, in several sites, including the oral cavity, gastrointestinal tract, and pancreatic tissue, along with the numerous small molecules and metabolites it produces, influences cancer progression and treatment by activating oncogenic signaling, enhancing oncogenic metabolic pathways, altering cancer cell proliferation, and triggering chronic inflammation that suppresses tumor immunity. Diagnostics and treatments based on or in combination with the microbiota offer novel insights to improve efficiency compared with existing therapies.

## 1. Introduction

The human microbiota is present at different sites on the surface and within the body, including the human skin, oropharynx, gastrointestinal (GI) tract, genitalia, and conjunctiva [1,2]. Microbiome research has been facilitated by the advent of next-generation sequencing techniques and either gas chromatography (GC-MS) or liquid chromatography (LC-MS), which allow for the assessment and analysis of microbiota composition, typically through biomarker gene sequencing [3,4]. Over the past 2 decades, numerous studies have revealed the effects of microbiome manipulation on different aspects of the physiology and metabolism of multicellular organisms, with implications for health and disease [1,5,6,7,8]. The GI microbiota, in particular, positively affects various host functions by producing numerous beneficial small molecules and metabolites, which play a significant role in individual physiology, inflammation, metabolism, immunity, nutrition, and neurology [9,10,11,12,13,14,15,16,17]. The disruption of the microbial host network has been implicated in many human pathological conditions, which probably influence cancer progression and treatment in both positive and negative ways [18,19,20,21,22].

Although cancer is generally considered to be a disease, caused by accumulating changes in the genome, epigenome, and environmental factors, microbes are associated with approximately 20% of human malignancies and are key mediators regulating cancer susceptibility and tumor progression [23,24,25]. The use of immune checkpoint inhibitors (ICIs) is increasingly recognized because of the critical functions of the microbiome in the dynamic regulation of immune homeostasis [26,27,28,29,30,31,32,33,34,35]. The microbiota modulates the tumor microenvironment (TME) locally through interactions with immune responses, hence, influencing cancer initiation, development, and treatment, and the effects range from harmful to beneficial [18,36,37]. The microbiota exerts vital functions on distant or proximal tumor tissues, primarily through three categories of microbial pathways, including manipulating the balance between host cell death and survival, activating immune system function, and responding to host-produced factors, ingested food, and drug metabolism, particularly in colorectal cancer, which is in close contact with the GI microbiota [18,38,39,40,41]. The GI tract, acting as the largest reservoir of microorganisms, plays a major role in metabolic health and immune regulation through multiple interactions with host cells [3,42,43].

Pancreatic cancer (PC), particularly pancreatic ductal adenocarcinoma (PDAC), the most common form of PC, with a dismal 5-year survival rate of approximately 10% at diagnosis, is an aggressive, devastating, and lethal form of human cancer [44,45,46,47,48]. Recent studies have described that certain microbiota contributes to cancer onset and progression by activating oncogenic signaling, enhancing oncogenic metabolic pathways, altering cancer cell proliferation, and triggering chronic inflammation that suppresses tumor immunity [42,49,50,51]. Results from patients with PDAC were compared with those from healthy controls, showing that the former may have different microbiomes in multiple body sites, including oral, GI, and pancreatic tissue [52,53,54,55]. Even at different stages of PDAC progression, the gut microbiota and metabolome of patients with resectable and unresectable PDAC showed differences [56]. Therefore, it can be inferred that the analysis and target of the microbiome may provide novel approaches with the potential to improve the efficacy of PDAC diagnosis and immunotherapy response [57].

This review will highlight recent studies analyzing the diverse microbiota affecting PC, elucidating the intricate cross-talk between the microbiota and various immune cells, thereby providing potential future therapeutic strategies to improve current treatment efficacy.

## 2. PC and the Microbiome

As shown in Figure 1, compared with healthy individuals, various microbiome changes are demonstrated in multiple body sites in patients with PC, including the oral, GI tract, and pancreas [52,55,58,59]. To achieve a better understanding of these microbiomes, clinical, demographic, epidemiological, and laboratory findings related to PC are under investigation, and oral rinses/swabs, saliva, blood, stool, biopsies, and tissue samples were examined for various microorganisms (Table 1). The detection of specimens is routinely achieved using several methods, including the analysis and size characterization of antibodies in the blood plasma, enzyme-linked immunosorbent assay, quantitative polymerase chain reaction (qPCR), 16S ribosomal ribonucleic acid (rRNA) gene sequencing, and microarrays. Recent data provided supportive evidence that conditions caused by oral microbiomes, such as periodontal disease or tooth loss, are related to the disease status of PC [60]. Correlations between the oral microbiota and PC carcinogenesis have been elucidated [61]. Several epidemiological and clinical studies have suggested an association between PC and Helicobacter pylori seropositivity [62,63,64]. To date, although molecular evidence remains scarce, a significant proportion of the clinically available literature proves the carcinogenesis of the hepatitis B virus (HBV) in the process of pancreatic tumorigenesis [65,66,67]. Research on the microbiome within the pancreas was also conducted. Based on multiple existing research findings, it can be hypothesized that PC may have some bacterial origins [49,58,68].

### 2.1. Oral Microbiota

The human oral cavity houses a diverse microbial community, harboring over 700 species of microorganisms. The microbiota that constitutes the oral microbiome remains relatively stable over time, indicating its vital impact on maintaining health. If oral disease states, including gingivitis, periodontal disease, and other diseases, happen, the ecological balance of the microbiota in the oral cavity will be altered [60,76,77,78,79]. Periodontitis, a severe and chronic oral infection leading to tooth loss and other health complications, which affects the supporting tissue of the teeth, including the gums, gingival tissue, and surrounding area, is thought to be associated with various forms of cancers, such as PC and colorectal cancer [53,69,78,80,81,82,83,84,85,86,87,88]. The PC risks linked to bad oral health, periodontal disease, pathogenic oral flora, and tooth loss have been fully vindicated and recognized as independent risk factors [78,80,81,89,90,91,92]. Therefore, the variation in taxa dominance and the diversity in microbial communities may either cause disease or reflect disease states [93,94,95]. Research conducted on both animal models and human subjects has demonstrated that means, including translocation and dissemination, result in the presence of the oral microbiota in the pancreas [53,96]. Preliminary studies have suggested that oral pathogens should be used as screening tests and potential biomarkers to be evaluated for the early diagnosis of PC [97]. Some specific microbiota in the pancreas are found to be similar to the oral microbiota [53,98,99,100,101,102,103]. Researchers believe that the dysbiosis of oral microbiota precedes the onset and progression of PC, instead of after tumorigenesis [104]. A considerable body of literature has investigated the key pathogens among the oral microbiota involved in the carcinogenesis of PC, specifically *Porphyromonas gingivalis* (*P. gingivalis*), *Fusobacterium*, *Neisseria elongata* (*N. elongata*), and *Streptococcus mitis* (*S. mitis*) [79,97].

#### 2.1.1. *P. gingivalis*

*P. gingivalis*, a kind of Gram-negative anaerobe, thrives in the development of chronic periodontitis [105]. When focusing on the interactions between the carriage of *P. gingivalis* and PC, the dose–response relationship is proven to be significant [54]. Collecting oral wash samples before diagnosis from both individuals with PC and healthy participants, a prospective nested case–control study investigated and characterized the oral microbiome and found that carriage of *P. gingivalis* is related to a 59% increased chance of subsequent PC infection [54]. *P. gingivalis* infection greatly contributes to the proliferation of the mouse cell line Panc02 along with the human PC cell lines PANC1 and MIA PaCa-2 [106]. To the highest degree, the concentration of plasma antibodies against *P. gingivalis* is associated with a 2-fold greater possibility of developing PC. As time goes by, the interrelation amplifies with a 5- or 7-year lag [53]. Notably, even several years before PC diagnosis, the risk of PC increases with the carriage of *P. gingivalis*, according to the concept that the spread of *P. gingivalis* from the oral cavity to the pancreatic tissue may promote the occurrence of cancer, regardless of how long it stays [54,106,107].

*P. gingivalis* can survive and persist with the host immune tissue by modifying the host’s immune response, for example, interacting with host receptors, altering signaling pathways, and invading host cells [108,109]. The challenge with live *P. gingivalis* reduced the secondary cytokine and chemokine responses of primary human gingival epithelial cells. The direct degradation of cytokines by *P. gingivalis* protease leads to the lack of secondary responses, and the degradation rates of IL-6 and IL-8 were apparently higher than those of IL-1β [108]. Researchers hypothesized that *P. gingivalis* can initiate inflammation [110]. Studies have suggested that patients with periodontal disease exhibit elevated biomarkers of systemic inflammation and evade immune responses associated with lipopolysaccharide (LPS) and Toll-like receptors (TLRs) [111]. The activation of TLRs has been shown to play an essential driving role in human PC, hence, proposing mechanistic interactions between microbial stimulation and PC [112].

#### 2.1.2. *Fusobacterium*

*Fusobacterium*, an anaerobic, Gram-negative oral bacterium, is indicated as a possible protective factor for PC, reducing its risk [54]. Notably, when present in PC tissue, it was associated with increased cancer-specific mortality [113]. Different studies have yielded opposite results. Several studies have found a significantly lower abundance of *Fusobacterium* in patients with PC, whereas previous studies have found a substantially higher abundance [79,114]. Prior research has shown that, in PC tissues of 283 samples, the *Fusobacterium* detection rate was 8.8%; however, the status of the tumor was not related to any molecular or clinical features. In contrast, according to a multivariate Cox regression analysis, the cancer-specific mortality of the *Fusobacterium*-positive group was much higher than that of the *Fusobacterium*-negative group. Until now, no outstanding correlation has been observed between the molecular alterations in PC and *Fusobacterium* species status [113]. Nevertheless, the existence of *Fusobacterium* was related to poor prognosis independently, which was consistent with the findings of case–control studies [79,113,115]. According to the study, bacterial profiles were associated with PC-related symptoms. In the study, compared with asymptomatic patients, symptomatic patients exhibited distinct bacterial profiles. For example, PC with bloating had higher abundances of *Porphyromonas*, *Fusobacterium*, and *Allobacterium* [114].

FadA, a surface adhesin, which plays an important role in cell adhesion and invasion, is acknowledged as being unique and highly conserved among *Fusobacterium* [116]. It is speculated that one of the molecular mechanisms causing PC may be FadA adhering to host epithelial cells [117]. FadA can invade host cells, either directly or pericellularly, via loosened cell–cell junctions. Vascular endothelial (VE)-cadherin, belonging to the cadherin family, acts as the endothelial receptor for FadA, and upon binding to it, VE-cadherin can translocate from the cell–cell junctions to intracellular compartments. As a result, the endothelial cell permeability is increased, allowing the bacteria to penetrate through loosened junctions [113,118]. However, questions regarding the role of *Fusobacterium* in pancreatic carcinogenesis remain to be addressed. As was recently reported in the literature, *Fusobacterium* increases the production of reactive oxygen species (ROS) and inflammatory cytokines, such as TNF and IL-6, in colorectal cancer; drives myeloid cell infiltration in intestinal tumors; and modulates the tumor immune microenvironment [70].

Though the results appear to be conflicting with those of other studies, the underlying mechanisms may lie in different observational conditions.

#### 2.1.3. *N. elongata* and *S. mitis*

The results of both the validation dataset of a retrospective study and a test cohort are consistent, showing that compared with controls, *N. elongata* and *S. mitis* in the saliva samples collected after PC diagnosis were lower [89]. The data were partially supported by the findings of Michaud et al., who showed a negative correlation of *S. mitis* antibodies with PC (but lack the data on *N. elongata*) [53]. Another result for the proportions of *N. elongata* was in the same direction in cases compared with controls as the former retrospective study; however, the results for *S. mitis* were not [89,119]. An ROC plot AUC value of 0.9 with 96.4% sensitivity and 82.1% specificity was concluded by distinguishing patients with PC from healthy individuals by combining the salivary RNA biomarkers of *N. elongata* and *S. mitis* [89].

#### 2.1.4. Others

Prior 16S sequencing studies emphasized the importance of the oral microbiota in PC progression and identified several potential microbial markers, for instance, *Aggregatibacter actinomycetemcomitans*, *N. elongata*, *P. gingivalis*, and *S. mitis* [54,89,97,117].

Compared with those in healthy subjects, concentrations of *Corynebacterium* and *Aggregobacteria* in patients with PC were lower; meanwhile, salivary RNA results showed that *Granulicatella adiacens* and *Bacteroides* were more common in those with PC [69,89,120]. Other than this, research recently found that *Aggregatibacter actinomycetemcomitans* was associated with a greater possibility of PC [54]. *Leptotrichia* is also proven to be a protective microorganism that reduces the danger of PC in a dose-dependent manner [54].

### 2.2. GI Microbiota

The number of microbial species contained in a healthy GI tract is as many as the cells making up the body and increases in density from the small intestine to the large intestine [121]. The microbiota aids in nutrient digesting, vitamin providing, infection preventing, and GI immune system shaping [122,123]. Recently, an increasing amount of data suggest that diseases outside the GI tract are also in control of the GI microbiota, such as metabolism, autoimmunity, and malignant liver diseases [124,125,126,127,128,129]. With increasing associations being discovered, clear and direct links between dysbiosis and disease status have been established [3,130,131,132,133]. A key instance is a link between the GI microbiome and colorectal cancer [131]. Because the pancreas is connected to the GI tract anatomically via the pancreatic duct system, evidently, the GI microbiota may affect the pancreas and vice versa. Specifically, a healthy pancreas shapes the GI microbiome and immune response, which controls pancreatic function and disease through altered immune responses [134]. Multiple studies have shown a clear association of distinct GI microbiome profiles with PC [52,135,136,137,138]. In disease states, alterations in the GI microbiota may be a cause or a consequence of the disease process. On the one hand, this effect could be conferred through the modulation of metabolites, such as short-chain fatty acids (SCFAs), or immune responses. Metabolomics analysis showed that altered metabolites, including amino acids, carnitine derivatives, lipids, and fatty acids, correlated with NF-kappa B signaling, the FXR/RXR pathway, etc. [56]. On the other hand, pancreatic factors, such as the excretion of antimicrobial drugs, may have a great impact on the constitution and functional properties of the GI microbiota [134,139].

In an animal experiment, the GI microbiota produced butyrate, a kind of SCFA, inducing the expression of cathelicidin-related antimicrobial peptide (CRAMP) in pancreatic beta cells [140]. Similarly, acetate, another SCFA from the GI microbiota, induces insulin secretion via the microbiome–brain β-cell axis [141]. Accordingly, the interplay between the GI microbiota and pancreatic factors is essential for both health and disease status. Microbial dysbiosis or imbalance can result in pancreatic dysfunction, even leading to diseases [142]. To summarize the recent studies, upper GI *Helicobacter pylori* (*H. pylori*) infection is one of the risk factors for PC [64,143]. *H. pylori* is supposed to influence carcinogenesis by promoting cell proliferation [144]. In addition to *H. pylori*, several studies have also shown the vital association between HBV and PC.

#### 2.2.1. *H. pylori*

*H. pylori*, a microbiota that infects the stomach, colonizes the gastric environment of 60.3% of individuals worldwide. Its prevalence is so high that it even exceeds 80% in areas with poor socioeconomic conditions [144]. *H. pylori* has been proven to lead to chronic active gastritis, which subsequently leads to a peptic ulcer and even gastric cancer. Usually, infections are accompanied by extragastric manifestations, such as immune thrombocytopenia or hypochromic anemia [145,146,147]. Meanwhile, *H. pylori* infection also accounts for other cardiovascular, neurological, allergic, metabolic, and hepatobiliary diseases [148,149].

Previous research has reported that in patients with PC, both the occurrence of *H. pylori* infection and the positive rate of *H. pylori* serum antibodies are higher [58]. *H. pylori* was found in the pancreatic tissue of patients with PC, chronic pancreatitis, multiple endocrine neoplasia type 1, and pancreatic neuroendocrine tumors. *H. pylori* DNA was detected in 75% of patients with PC, whereas all specimens from the normal pancreas and other benign pancreatic diseases were negative [150]. As part of the dysbiosis of the microbiota, *H. pylori* has been recognized as the potential trigger of autoimmune inflammation in the pancreas.

*H. pylori* colonizes the gastric mucosa in two ways. One way is primarily linked with antral gastritis, resulting in increased gastrin production, possibly through local damage, the release of somatostatin, and a decrease in antral D cells. This leads to excessive acid secretion, increasing the susceptibility to prepyloric and duodenal ulcers. Therefore, this colonized form of *H. pylori* was identified as a dangerous factor for PC development [144]. The colonization of *H. pylori* is related to the activation of molecular pathways associated with PC initiation and maturation, thereby contributing to PC malignancy [151].

*H. pylori* is considered to be indirectly involved in the onset and progression of PC [152]. Via PCR, the expression of *H. pylori* DNA in pancreatic juice or tissue could not be detected in chronic pancreatitis and PC, indicating that *H. pylori* indirectly triggers the emergence of PC, consistent with other studies. The possible indirect mechanisms are immune escapes and inflammatory responses [153]. *H. pylori* can secrete cytotoxin-associated proteins and vacuolar proteins, promoting chronic inflammatory oxidative stress and damaging host DNA to trigger cellular carcinogenesis [72]. According to a prospective cohort study involving 51,529 male subjects, the increased risk of PC associated with gastric ulcers may be due to greater inflammatory responses and endogenous nitrosation because of *H. pylori* infection [135].

#### 2.2.2. HBV and Hepatitis C Virus (HCV)

Both HBV and HCV are liver-tropic pathogens and have well-known carcinogenic properties [154]. Nevertheless, HBV or HCV infection is not limited to the liver, which has also been discovered in extrahepatic tissues, including the pancreas [155,156]. This might contribute to the undeniable role of these two viruses in both the occurrence and development of extrahepatic malignancies, such as PC [157,158]. The correlation between PC and HBV infection has been summarized by multiple meta-analyses [159,160,161,162,163,164,165,166]. According to HBV carrier status, most reviews have reported a positive association, with relative risks ranging from 1.2 to 3.8. The findings of two meta-analyses have shown that HCV infection is positively correlated with PC risk [167,168].

Specifically, HBcAg and HBsAg have been detected in the cytoplasm of pancreatic acinar cells. Patients infected with chronic HBV were partially associated with elevated serum and urinary pancreatic enzyme levels [169]. HBsAg was also found in the pancreatic juice of HBV-infected patients and was related to the progression of chronic pancreatitis, suggesting that HBV-related pancreatitis is a precursor to PC [170,171]. Moreover, viruses, including HBV and HCV, may contribute to the progression of PC by activating the inflammation process and regulating the PI3K/AKT signaling pathway [155].

#### 2.2.3. Others

Compared with that in healthy controls, the abundance of microbiota belonging to the phyla *Firmicutes* [49], *Bacteroidetes* [49], *Proteobacteria* [49], *Actinobacteria*, *Fusobacteria*, and *Verrucomicrobia* [172] and the genera *Porphyromonas*, *Bifidobacterium* [115], *Prevotella* and *Synergistetes*, along with the archaeal phylum *Euryarchaeota* [49], has been proven to increase significantly in PC. Meanwhile, decreased abundances of other GI microbiota in PC, including *Firmicutes* [136], *Proteobacteria* [52], and *Lactobacillus* [115], have been reported. In patients with PC, it was noted that the abundance of beneficial probiotics and butyrate-producing bacteria decreased, whereas the abundance of potentially pathogenic LPS-producing bacteria increased [52]. The duodenal mucosa of patients with PC is more made up of *Acinetobacter*, *Sphingobium*, *Deinococcus*, *Delftia*, *Massilia*, *Rahnella*, *Oceanobacillus*, and *Aquabacterium* [173]. Aykut et al. amplified the ITS1 region of the 18S rRNA gene to test fecal and tumor fungal communities in patients with PC. The results showed that in the intestine and tumor tissues, the most common phyla were *Ascomycota* and *Basidiomycota*. The population of fungi (mainly *Malassezia* species) detected in PC tissues was 3000-fold higher than that in a normal pancreas [68]. According to research, *Veroella*, *Enterococcus*, *Shigella*, *Streptococcus*, and *Enterobacter* are considered to be the five most crucial genera in bile [173]. However, through genetic sequence analysis, another study involving patients with PC investigated the presence of microbiota in bile samples, finding that the most predominant microbes were *Enterobacter* and *Enterococcus* spp. [138]. The aforementioned microbiota was reported to be strongly associated with the onset and progression of PC, which makes it possible as a biomarker for the noninvasive diagnosis of this disease [174].

### 2.3. Pancreatic Microbiota

For a long time, the pancreas was considered a sterile organ, but only recently has the microbiota been identified in pancreatic tumor tissue and cyst fluid, called the PC intratumoral microbiota [175,176,177]. Via bacterial 16S rRNA gene-specific PCR, the analysis of the microbial composition of pancreatic cyst fluids revealed a predominance of *Bacteroides*, *Escherichia*/*Shigella*, and *Acidaminococcus* [177]. In pancreatic cancer tissue, Pushalkar et al. identified that *Proteobacteria*, *Bacteroidetes*, and *Firmicutes* were more enriched [49]. According to preclinical models, a 1000-fold higher microbial abundance was detected in pancreatic tumor tissues than in healthy pancreatic tissue [49]. Species belonging to the *Firmicutes* and *Proteobacteria* phyla accounted for most of the bacterial sequences found in PC, similar to the composition of a healthy intestinal microbiome [49,178]. *Gamma-proteobacteria* were detected in gemcitabine-resistant PC tissue samples, and researchers speculate whether the microbiota participates in tumor pathogenesis or exists coincidentally—they might play a key part in mediating resistance to chemotherapy [179]. Longitudinal analysis of age-matched KC (p48Cre; LSL-KrasG12D) and wild-type mice revealed that in KC mice, specific microbiota was enriched, and the amplest species was *Bifidobacterium pseudolongum* [58]. Aykut et al. found that PC samples from humans were obviously enriched with *Malassezia* spp. [68]. Several studies found that patients with PC have a higher intrapancreatic abundance of oral *Fusobacterium* than non-cancer controls, which is independently associated with reduced patient survival [113,115]. Because of its microbial environment, the pancreas is not sterile, leading to the occurrence and development of PC being affected [180]. The role of intratumor microbes in PC progression and the modulation of response to immunotherapy has been investigated [49]. The pancreatic microbiota was further proven to accurately predict long-term survivorship in patients with PC. Experimental evidence also indicated that the long-term survivorship of patients with PC was modified by the GI microbiome [181,182].

## 3. Microbiome and Immunity

Some pieces of evidence gradually confirmed that the microbiota can interact with the host to regulate antitumor immunity to shape cancer development and to impact the response to tumor therapy, particularly ICIs [26,27,28,183,184]. Indeed, in the cancer context, by binding to inherent and acquired immune cells, the microbiota exerts an immune effect locally and systemically, remodeling the immunity of the TME [185]. Notably, one of the major ways in which the microbiota modulates antitumor immunity is through metabolites, that is, small molecules that can diffuse and influence antitumor immune responses locally and systemically to enhance ICI efficiency [186,187]. Studies have confirmed that the microbiota can regulate dendritic cells (DCs) [188], monocytes/macrophages [189], natural killer cells [190], CD8+ T cells [188], and CD4+ T cells [191], among others, to stimulate antitumor immune responses.

### 3.1. Innate Immunity

#### 3.1.1. DCs

DCs, a diverse group of specialized antigen-presenting cells, are important in antitumor immunity and T-cell activation. The origination of conventional DCs (cDCs) is a common DC precursor in the bone marrow, which has been considered to be a critical mediator of antigen-priming and T-cell activity [192,193]. For example, the number and function of cDCs can determine whether adaptive immune responses to tumor neoantigens are protective or deleterious [194]. Microbiota antigens or metabolites with immunomodulators can mobilize and activate DCs to reverse immune tolerance induced by immature DCs [50,195]. By contributing to DC maturation and IL-12-dependent Th1 cellular immunity, *Bacteroides fragilis* enhances the antitumor ability of CTLA-4 blockade [74,196]. Oral antibiotics lead to antitumor immune activation and the suppression of the tumor burden in PC-bearing mouse models [197].

In adult mice, the GI microbiota controls the production of CRAMP via pancreatic endocrine cells through SCFAs. CRAMP has a positive immunoregulatory impact on pancreatic macrophages and cDCs and maintains immune homeostasis in the tissue by inducing regulatory T cells (Tregs) [140]. Eleven microbiota strongly induced CD8+ T cells, which produce interferon-γ (IFNγ) in the gut, acting together on both CD103+ DCs and major histocompatibility (MHC) class Ia molecules, aimed at tumor growth inhibition [188]. Gut dysbiosis was found to mediate experimental autoimmune pancreatitis (AIP) by activating pDCs, subsequently producing large amounts of IFN-α and IL-33 [198]. Upon activation, the costimulatory receptor CD40 can license DCs to re-educate macrophages to a tumoricidal phenotype and reverse tumor-associated fibrosis, which may enhance the chemotherapeutic efficacy in PC [199,200].

#### 3.1.2. Monocytes/Macrophages

Monocytes and macrophages are essential inherent immune effector cells in maintaining homeostasis [201]. Macrophages recruited to the TME adopt an immunosuppressive, proangiogenic state and prevent CD4+ T cells from entering the TME, thereby supporting PC progression [202]. In PC xenografts from microbiota-depleted mice, CD45+ cells statistically significantly increase, suggesting that the observed phenotype is related to the suppression of the inherent immunity mediated by the microbiota [128]. LPS, a major outer-surface membrane component of Gram-negative bacteria, such as *P. gingivalis*, is recognized by TLR4 on host innate immune cells, modulating NF-κB via pathways both dependent and independent of MyD88, thus leading to the release of proinflammatory cytokines [69,203]. Mouse model studies have found that the TLR4/MyD88 pathway may be associated with inflammation and PC progression. Specifically, LPS was reported to promote PC as blocked MyD88-dependent pathways (via DC-mediated TH2 deviation), whereas blocking TLR4 (via TRIF) and MyD88-independent pathways in the same pathway (via TRIF) can prevent PC [204]. TLRs appear to contribute to PC development, specifically expressed at high levels in human PC tissues rather than in the normal pancreas [205]. Microbiota metabolites may be recognized by TLRs or may stimulate inflammasome-mediated cytokine secretion [206]. For example, the intratumoral microbiota in mouse and human PC promotes carcinogenesis by inducing a tolerogenic immune program, including the inhibition of monocyte differentiation by selective TLRs and T-cell anergy [10,49]. A bacterial ablation orthotopic PC mouse model prevents PC from invasion by remodeling the TME, including reducing MDSC numbers, polarizing macrophages to an M1 phenotype, promoting Th1 differentiation, and activating CD8+ T cells. Mechanistically, the PC microbiome increases the sensitivity of immune surveillance and immunotherapy by differentially activating selected TLRs in monocytes, including TLR2 and TLR5 [42,49,139,207]. TLR activation can induce the STAT3 and NF-κB pathways, acting as carcinogenic factors to increase cell proliferation and to inhibit apoptosis [208].

Furthermore, immunohistochemical staining analysis showed a close correlation between tumor-associated macrophages and nerve density in PC tissues, suggesting that there is an interaction of paracrine signaling between nerves and macrophages [209,210]. Other than neurotransmitters, enteric neurons also release growth factors, such as CSF1, to interact with macrophages in healthy tissues, indicating that cytokines can mediate the neuroimmune cross-talk in PC [211].

#### 3.1.3. NK Cells

NK cells, a type of cytotoxic lymphocyte critical to the innate immune system, can kill viral infections and cancer cells [212]. NK cells are responsible for monitoring circulating tumor cells and preventing tumor cell metastasis. If NK cells are depleted or inhibited, tumor growth and escape may occur [213,214].

Studies have reported that NK cells modulate the abundance of DC and CD8+ T cells in the TME and affect the response to ICIs [212,213,215,216]. By suppressing the immunity of CD8+ T cells, NK cells are activated to promote immunopathology and chronic infection [217]. The downregulation of activating receptor expression and impaired function of NK cells are common immune evasion mechanisms and have been implicated in PC development [218,219,220,221,222,223]. Recently, an increasing number of researchers have identified the interactions between NK cells and the microbiota. Under the effect of PD-1 blockade, patients with non-small-cell lung cancer (NSCLC) with high microbial diversity showed a higher abundance of peripheral-specific memory CD8+ T-cell and NK cell subsets [224]. The interaction mechanism between *Bifidobacteria* and NK cells was demonstrated to be mediated by hippurate and under high-salt-diet (HSD)-mediated tumor immunity. An HSD increases the abundance of *Bifidobacteria* and promotes intestinal permeability, leading to the intratumoral localization of *Bifidobacteria*, which enhances NK cell function to induce antitumor immunity [190]. A randomized controlled meta-analysis has shown that dietary supplementation of *Bifidobacteria* probiotics directly enhances NK cell function in the elderly [225].

### 3.2. Adaptive Immunity

#### 3.2.1. CD4+ T Cells

CD4+ T cells are crucial in regulating immune responses by signaling other types of immune cells [226]. Naïve CD4+ T cells can be differentiated into four types, namely helper T cells (i.e., Th1, Th2, and Th17) and Tregs, which are involved in the tumor immune microenvironment, tumor immune escape, immune homeostasis, and antitumor immunity [227,228,229]. Patients with PC show a disorder of the Th17/Treg balance, with lower Th17 cells and higher Tregs. It mainly affects the expression of the cytokines IL-10, IL-23, INF-γ, TGF-β, and IL-17 by modulating transcription factors, such as RORα, RORγt, FoxP3, and CTLA-4 [230]. Th2 cells infiltrate the pancreas and secrete type 2 cytokines (i.e., IL-4 and IL-13) early in tumorigenesis, promoting metabolic reprogramming and cancer cell proliferation in mice with KrasG12D-driven PCs. Similar to type 2 immune responses driving PC development in mouse models, patients with PC with a higher infiltration of Th1 (CD45+CD3+CD4+Tbet+) cells demonstrate a higher level of survival than those with predominantly TH2 (CD45+CD3+CD4+Gata3+)-polarized lymphoid cell infiltration [231]. Furthermore, circulating levels of IL-4 were inversely correlated with disease-free survival (DFS) in individuals with PC [232].

Mouse model experiments have shown that *B. pseudolongum* promotes Th1 transcriptional differentiation and antitumor immunity mainly through the GI microbial metabolite inosine [75]. *Bacteroides fragilis* promotes the mobilization of lamina propria DCs to stimulate IL-12-dependent Th1 immune responses [74]. *Faecalibacterium* increases the proportion of CD4+ T cells and serum CD25 production and decreases the proportion of Tregs in peripheral blood in human patients, thereby inducing long-term clinical benefit of ipilimumab, an antibody against cytotoxic T-lymphocyte-associated antigen 4 (CTLA-4) [74,233].

#### 3.2.2. CD8+ T Cells

CD8+ T cells are pivotal players in anticancer immunity, releasing IFNγ and TNFα to eliminate cancer cells [206]. In mouse models, mice with PC-enriched CD8+ T cells survive longer [234]. In patients with PC, tissue infiltrated by CD8+ T cells also showed longer survival [235]. Specific microbiota induces CD8+ T cells in the systemic circulation or TME [50]. For instance, patients with melanoma with higher relative abundances of favorable microbiota, including *Clostridium*, *Ruminococcus*, and *Faecalibacterium*, demonstrated increased antigen presentation along with improved function of effector CD4+ and CD8+ T cells in the peripheral blood and TME to improve the antitumor efficacy of ICIs [236].

T-cell immunity is a well-established element for long-term PC survival. Prior research has reported that long-term-survival (LTS) patients with PDAC showed high levels of CD8+ T-cell tumor infiltration, Th1-related gene expression, and M1 macrophage differentiation [49,181,237,238]. Through human-to-mice fecal microbiota transplantation (FMT) experiments involving patients with PC with short survival (STS), patients who had PC resected more than 5 years previously showed no evidence of disease (LTS-NED), and in healthy controls (HC), research demonstrated that the favorable impact of LTS-NED-associated GI/tumor microbiota is mediated by CD8+ T cells [181]. The SCFAs secreted by beneficial commensal microbiota can promote the antitumor responses of CD8+ T cells [239]. Evidence from clinical trials suggests that *Actinobacteria* and *Firmicutes* are enriched in FMT and PD-1 blocking reactants. A combined blockade of FMT and PD-1 stimulates mucosal-associated invariant T cells and CD56+CD8+ T cells in peripheral blood mononuclear cells and upregulates human leukocyte antigen class II genes CD74 and GZMK T cells at the tumor site expressing CD8+ T cells [183]. Meanwhile, the use of FMT combined with PD-1 blockade increased the relative abundance of Enterococcus in refractory metastatic melanoma, increasing intratumoral CD8+ T-cell infiltration and tumor cell necrosis [240].

## 4. Clinical Trials

Despite tremendous efforts and growing evidence from research and clinical trials related to PC, the 5-year overall survival rate for patients has only increased marginally from 5% to 9% [241]. This small advancement has been achieved mostly through recent improvements in neoadjuvant and adjuvant treatment strategies and perioperative care. While new treatments, including immunotherapy, have significantly improved the prognosis of patients with cancer based on clinical data, there is heterogeneity in the overall outcomes, and existing biomarkers cannot reliably and accurately predict the prognosis of these patients. Currently, the prediction of cancer treatment response has focused on tumor-intrinsic characteristics. The discovery of other relevant risk factors, the identification of appropriate biomarkers, and a better understanding of the major players that influence PC treatment outcomes are important directions for future research [242]. Nowadays, immunotherapy approaches for PC under investigation include ICIs, adoptive cell therapy, immune agonist therapy, cancer vaccine, bone-marrow-targeted therapy, and combinations of chemoradiotherapy and other molecularly targeted agents [51,243].

Immune checkpoint molecules, negative regulatory molecules of immune responses to avoid immune injury, are important in maintaining self-tolerance, preventing autoimmune responses, and minimizing tissue damage by keeping the activation of the immune system within the normal range. ICIs, monoclonal antibodies that antagonize immunosuppressive pathways known as checkpoints, have recently emerged as a promising idea for new cancer treatments [244,245]. Common ICIs include nivolumab and pembrolizumab (targeting programmed cell death protein-1 (PD-1)); atezolizumab, avelumab, and durvalumab (targeting programmed cell death protein ligand-1, (PD-L1)); and ipilimumab and tremelimumab (targeting cytotoxic T-lymphocyte–associated antigen 4, (CTLA-4)), revolutionizing cancer care over the past decade [246].

Unfortunately, although ICI therapy can stimulate the activation of T cells and potent antitumor immune responses, it can also bring about severe inflammatory side effects—immune-related adverse effects (IrAEs). While killing tumor cells, ICI therapy disrupts the immune balance of the host, potentially leading to immune-related pneumonia, immune-related colitis, and even life-threatening immune-related myocarditis [50]. IrAEs most commonly involve the GI tract, endocrine glands, skin, and liver [247]. The effects may occur in up to 90% of patients treated with anti-CTLA-4 antibodies and in 70% of patients treated with PD-1/PD-L1 antibodies [247,248].

There is no doubt that the microbiota is one of the parameters that modulates the immune setpoint of cancer [249]. The microbiome is increasingly recognized for its impact on host immunity and favorable effects on cancer therapy. Importantly, strategies to alter the microbiome could provide new perspectives for cancer treatment to improve outcomes [28]. A strong correlation has been reported between the microbiome and response to ICIs [191]. The GI microbiota may be a potential factor influencing the success of checkpoint blockade immunotherapy [250]. Comprehending the biological mechanisms of the microbiota along with metabolites in response to antitumor immunity and immunotherapy is critical for manipulating microbial activity rationally to improve the efficacy of ICI therapy [251]. Table 2 shows that the therapeutic strategies combining the microbiota with ICIs, including appropriate antibiotic selection, probiotic intake, FMT, and bacterial genetic engineering, can bring hope to patients with PC [50,252,253,254].

Lately, the microbiome was reported to improve the antitumor efficacy of PD-1 and CTLA4 blockade therapies [74]. The GI microbiota has been shown to increase the efficacy of the blockade of the PD-1 protein and its ligand PD-L1. Most tumor responses were durable beyond 1 year [73,247,248,256,257]. Experimental results demonstrated that ICI therapy and adoptive cell therapy, which use tumor-specific CD8+ CTLs, are influenced by the composition of the GI microbiome [258,259]. The GI microbiota can produce SCFAs, such as *Eubacteria*, *Lactobacilli*, and *Streptococci*, which were positively associated with anti-PD-1/PD-L1 responses in various types of GI cancer [255]. *Eubacterium* fermentates fiber into SCFAs, while propionic acid, *Lactobacillus*, and *Streptococcus* mainly contribute to lactic acid production in the gut. If these genera in responders are relatively abundant, the gut environment may have beneficial immune activity [255]. *Bifidobacterium* contributed to enhancing antitumor immunity and PD-L1 blockade therapy efficacy. The results have proven that oral *Bifidobacterium* alone improved the extent of tumor control similar to PD-L1-specific antibody treatment (checkpoint blockade), and the combination treatment nearly eliminated tumorigenesis. It is the enhanced DC function that mediates the effect, enhancing CD8+ T-cell priming and accumulation in the TME [73]. A correlation between the microbiome and immunotherapy has also been reported, showing that antimicrobial therapy and αPD-1 therapy have a synergistic effect on decreasing tumor size in vivo [49]. Additionally, *Bifidobacterium pseudolongum*, *Lactobacillus johnsonii*, and *Olsenella* were found to significantly enhance the efficacy of ICIs in vivo [75].

## 5. Conclusions

Tumor formation in PC is complicated and involves multiple microbial compartments, including different sites [260]. For example, oral pathogens travel through the gut or other ways to the pancreas, bringing about inflammation that can lead to cancer [142].

The microbiota influences PC development in ways that affect immunity, including innate and adaptive immunity. Metabolites of dysregulated microbiota also influence tumorigenesis in pro-cancer or anticancer ways, potentially inducing and maintaining an inflammatory state and affecting oncogenic and cellular signaling pathways. The microbiome can be a predictive biomarker of treatment efficacy and safety, helping clinical physicians to better monitor the disease progression; it can also be combined with other therapies to improve treatment outcomes; direct manipulation of certain immune-stimulating metabolites or compounds from the microbiome could also help improve the precision of therapeutic strategies. Therefore, studies on the microbiome may enhance the effectiveness and robustness of personalized and precision medicine.

The microbiome offers patients with PC more opportunities for novel therapeutic targets. However, this research area is still in its infancy. Currently, in the PC status, the linkage between the microbiota and cancer, as well as the antitumor mechanism of the microbiota, has not been elucidated. The role of the microbiota, not only in the pathogenesis but also in the progression of the disease, will be the most important consideration. The microbiota has a multifaceted impact on tumor ICI therapy, ranging from facilitating to hindering. The results of the analysis of PC microbiota in existing studies are contradictory to some extent. Larger and more comprehensive studies are required to clarify the correlation and causality more clearly. Regardless of whether it is favorable or not, microbiome signatures targeting immune cells or pathways should be accurately classified, and their roles in different TMEs should be well understood. Research combining microbiology, immunology, metabolomics, molecular pathology, tumor genomics, and multiple dimensions will help assess the disease relations, along with the diagnostic and therapeutic potential of the microbiota, thereby forming a huge picture of PC formation and development.

## Figures and Tables

**Figure 1 microorganisms-11-01240-f001:**
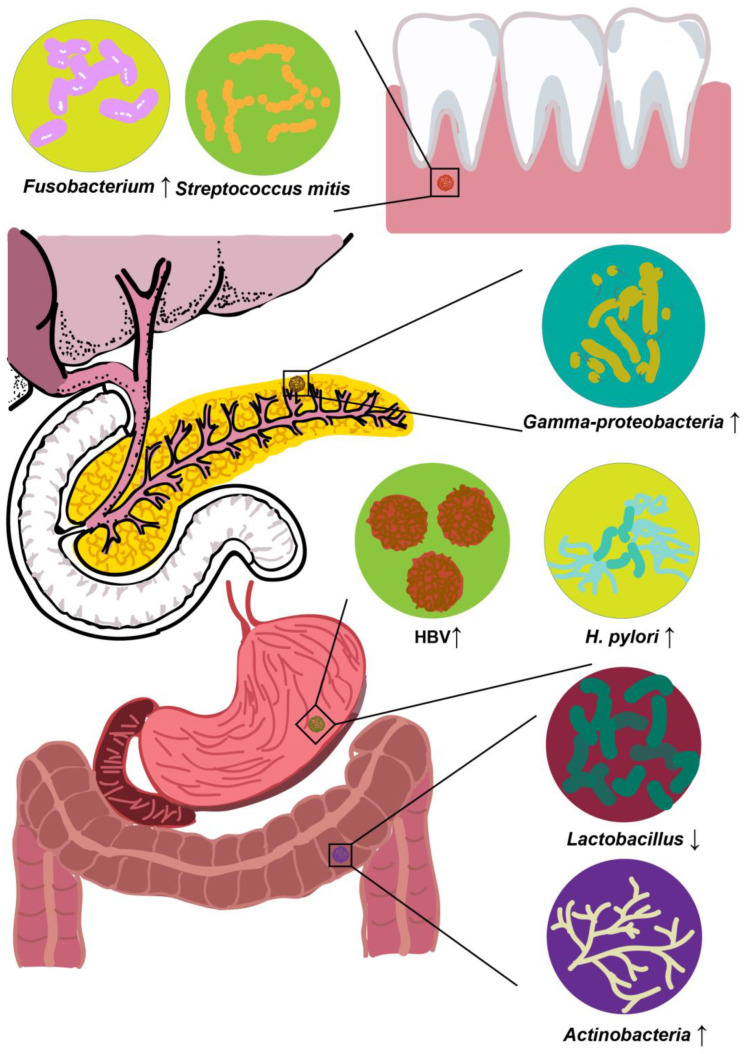
Specific microbiota associated with PC, whose abundances increase, decrease, or we are unsure. *Fusobacterium* and *Streptococcus mitis* are typical oral microbiota related to PC. *Gamma-proteobacteria* were the predominant bacteria that were first identified in human pancreatic cancer tissues. The gastrointestinal microbiota, including *H. pylori* and HBV, were demonstrated to be involved in PC. Notably, *Lactobacillus* and *Actinobacteria* exist in the colon, demonstrated to be related to PC.

**Table 1 microorganisms-11-01240-t001:** Studies on the microbiota that target innate and adaptive immune cells.

Authors	Year	Body Sites	Microbiota	Biological Effects	Ref.	PMID
P. M. Bracci	2017	Oral cavity	*P. gingivalis*	triggers innate immune response: recognizes TLR4, stimulates MyD88-dependent and MyD88-independent pathways, activates the NF-κB pathway, release proinflammatory cytokines	[69]	29189325
A. D. Kostic et al.	2013	Oral cavity	*Fusobacterium*	increases production of ROS and inflammatory cytokines (e.g., IL-6 and TNF)	[70]	23954159
M. Uribe-Herranz et al.	2018	GI tract	*Bacteroidales*	enhances the activity of antitumor-specific effector T cells, increases the levels of cDC and IL-12	[71]	29467322
J. Knorr et al.	2019	GI tract	*H. pylori*	secretes cytotoxin-associated proteins and vacuolar proteins that promote chronic inflammatory oxidative stress and damage host DNA to trigger cellular carcinogenesis	[72]	31130493
A. Sivan et al.	2015	GI tract	*Bifidobacterium*	promotes T-cell mediated antitumor immunity	[73]	26541606
M. Vetizou et al.	2015	GI tract	*Bacteroides fragilis*	promotes the mobilization of lamina propria DCs to stimulate IL-12-dependent Th1 immune responses	[74]	26541610
L. F. Mager et al.	2020	Pancreatic tissue	*B. pseudolongum*	a purine metabolite (Inosine), directly binds to and inhibits the ubiquitin-activating enzyme UBA6, enhancing tumor-intrinsic immunogenicity, thereby sensitizing tumor cells to T-cell-mediated cytotoxicity	[75]	32792462

**Table 2 microorganisms-11-01240-t002:** Representative clinical trials of microbiota-linked cancer, especially including ICIs.

Researchers	Year	Microbiota	Effects	Ref.	PMID
Z. Peng et al.	2020	*Eubacterium*, *Lactobacillus*, and *Streptococcus*	Positively associated with anti-PD-1/PD-L1 response	[255]	32855157
A. Sivan et al.	2015	*Bifidobacterium*	oral *Bifidobacterium* alone improves tumor control to the same extent as ICIs, and the combination treatment nearly eliminated tumor growth	[73]	26541606
L. F. Mager et al.	2020	*Bifidobacterium pseudolongum*, *Lactobacillus johnsonii*, and *Olsenella species*	Significantly enhances efficacy of ICIs	[75]	32792462
N. Chaput et al.	2017	*Faecalibacterium*	Increases the proportion of CD4+ T cells and serum CD25 production, decreases the proportion of Treg cells in peripheral blood in human patients, thereby inducing long-term clinical benefit of ipilimumab, an antibody against CTLA-4	[233]	28368458

## Data Availability

The data presented in this study are available in the review.

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
