# Peer review of "Microbiota Regulates Pancreatic Cancer Carcinogenesis through Altered Immune Response"

_microorganisms, 2023, doi:10.3390/microorganisms11051240_

Round 1

Reviewer 1 Report

The review titled "Microbiota regulates pancreatic cancer carcinogenesis through altered immune responseby Chai Y et al provides an overview of the current knowledge on the topic. In general, the authors made a thorough and critical analysis of the literature. To improve the present work, some minor points may be addressed: 

-        English check. Correct and rephrase several sentences by a native speaker. 

-        Be coherent with the number of citations. Having several citations for one single statement is not recommended (for example, section 3, “Previous researches have confirmed that the microbiota can regulate dendritic cells [217, 218, 226-229]”). Reduce the number of citations, focus on original articles rather than reviews, and keep the most relevant references that correspond with the facts stated. Likewise, if authors write that there are “several studies” (for example in section 2, Ref 63)”, more than 1 citation must be referred to.

-        Unlike the other sections, the Abstract, Introduction, and Conclusion need to be highly improved. Remove any irrelevant information or long listing and try to have a more critical and topic-related text. 

-        Several of the abbreviations must be fixed. Please follow good writing practice and keep consistency throughout the text. 

-        Figure 1. The colon harbours most of the microbiota, however, it is poorly represented in the figure. Increase the value of the figure by adding the intestine and more specific microbiota related to PC. If required, specify if it is increased or decreased. Moreover, the figure legend has to be improved and further describe the image. 

-        Neither the figure nor the tables are cited in the text. They must be mentioned in the corresponding sections. 

Author Response

Response to Review

We would like to thank you for your thoughtful comments, which have greatly improved our work. We have revised the manuscript according to your comments and suggestions. In the following, the reviewers’ comments appear in black, our point-by-point responses and the changes made to the manuscript are marked in blue.

 Reviewer #1:

Q1: English check. Correct and rephrase several sentences by a native speaker.

ANS: Thank you for your effort to carefully review our manuscript. The manuscript has undergone extensive English revisions accordingly.

Q2: Be coherent with the number of citations. Having several citations for one single statement is not recommended (for example, section 3, “Previous researches have confirmed that the microbiota can regulate dendritic cells [217, 218, 226-229]”). Reduce the number of citations, focus on original articles rather than reviews, and keep the most relevant references that correspond with the facts stated. Likewise, if authors write that there are “several studies” (for example in section 2, Ref 63)”, more than 1 citation must be referred to.

ANS: Thanks for your reminding. We revised the citations accordingly, especially those you have mentioned.

Q3: Unlike the other sections, the Abstract, Introduction, and Conclusion need to be highly improved. Remove any irrelevant information or long listing and try to have a more critical and topic-related text.

ANS: Thanks for your suggestion. The manuscript has been revised accordingly.

Q4: Several of the abbreviations must be fixed. Please follow good writing practice and keep consistency throughout the text.

ANS: Thank you for your thoughtful advice. Several abbreviations have been fixed in the revised manuscript, including Pancreatic cancer (PC), gastrointestinal (GI), tumor microenvironment (TME), and immune checkpoint inhibitors (ICIs). The manuscript has been revised accordingly.

Q5: Figure 1. The colon harbours most of the microbiota, however, it is poorly represented in the figure. Increase the value of the figure by adding the intestine and more specific microbiota related to PC. If required, specify if it is increased or decreased. Moreover, the figure legend has to be improved and further describe the image.

ANS: Thank you for your kind advice. The intestine and some specific microbiota involved in PC have been added; The changes in microbiota that most studies agree on were specified. A detailed description was also added to the figure legend. The manuscript has been revised accordingly.

Q6: Neither the figure nor the tables are cited in the text. They must be mentioned in the corresponding sections.

ANS: Thanks for your helpful advice. The manuscript has been revised accordingly.

Reviewer 2 Report

Interesting review on the  possible pathophysiological link between gut and oral microbiota and pancreatic carcinogenesis.

points of criticism:

1. Authors should add more data on the role of microbial metabolites involved in the pancreatic carcinogenesis. The recent publication by Guo X et al showed that patients with resectable pancreatic ductal adenocarcinoma (PDAC) show different gut microbiota compositions and metabolomic  profiles  as compared to patients wit resectable PDAC. Function and network analyses showed that altered metabolites are linked to NFkappa B signaling, FXR/RXR signaling etc. Please include this information in the present review.

Please comment also on intratumoral microbiota that represents an important force in treating pancreatic cancer.

Author Response

Reviewer #2:

Q1: Authors should add more data on the role of microbial metabolites involved in the pancreatic carcinogenesis. The recent publication by Guo X et al showed that patients with resectable pancreatic ductal adenocarcinoma (PDAC) show different gut microbiota compositions and metabolomic  profiles  as compared to patients wit resectable PDAC. Function and network analyses showed that altered metabolites are linked to NFkappa B signaling, FXR/RXR signaling etc. Please include this information in the present review.

ANS: Thank you for your constructive advice. According to the recent publication by Guo X et al., we add the information in this review as followed. “Even at different stages of PDAC progression, the gut microbiota and metabolome of patients with resectable and unresectable PDAC showed differences [56].” “Metabolomics analysis showed that altered metabolites including amino acids, carnitine derivatives, lipids, and fatty acids correlated with NF-kappa B signaling, the FXR/RXR pathway, etc.[56].” The manuscript has been revised accordingly.

Q2: Please comment also on intratumoral microbiota that represents an important force in treating pancreatic cancer.

ANS: Thank you for your thoughtful advice. Comments on intratumoral microbiota have been added to the section 2.3. The manuscript has been revised accordingly.